# A new method to quantify particulate sodium and potassium salts (nitrate, chloride, and sulfate) by thermal desorption aerosol mass spectrometry

Yuya Kobayashi and Nobuyuki Takegawa

Department of Chemistry, Graduate School of Science, Tokyo Metropolitan University, Hachioji, Tokyo 192-0397, Japan

*Correspondence to*: Nobuyuki Takegawa (takegawa@tmu.ac.jp)

**Abstract.** The reaction of sea salt (or biomass burning) particles with sulfuric acid and nitric acid leads to the displacement of chloride relative to sodium (or potassium). We have developed a new particle mass spectrometer to quantify non-refractory and refractory sulfate aerosols (referred to as refractory aerosol thermal desorption mass spectrometer: rTDMS). The combination of a graphite particle collector and a carbon dioxide laser enables high desorption temperature (blackbody equivalent radiation temperature of up to 930°C). Ion signals originating from evolved gas molecules are detected by a quadrupole mass spectrometer. Here we propose a new method to quantify the mass concentrations of sodium nitrate ($NaNO_3$: SN), sodium chloride (NaCl: SC), sodium sulfate ($Na_2SO_4$: SS), potassium nitrate ($KNO_3$: PN), potassium chloride (KCl: PC), and potassium sulfate ($K_2SO_4$: PS) particles by using the rTDMS. Laboratory experiments were performed to test the sensitivities of the rTDMS to various types of particles. We measured ion signals originating from single-component particles for each compound, and found a good linearity ($r^2 > 0.8$) between the major ion signals and mass loadings. We also measured ion signals originating from internally mixed SN + SC + SS (or PN + PC + PS) particles, and found that the temporal profiles of ion signals at $m/z$ 23 ($Na^+$) (or 39; $K^+$) were characterized by three sequential peaks associated with the evolution of the desorption temperature. We tested potential interferences in the quantification of sea salt particles under real-world conditions by artificially generating "modified" sea salt particles from a mixture of diluted seawater and SN (or SS) solution. The SS/SC ratios estimated from the ion signals at $m/z$ 23, 36 ($H^{35}Cl^+$), and 48 ($SO^+$) agreed well with those predicted from the solution concentrations to within ~10%. The SN/SC ratios estimated from the ion signals at $m/z$ 30 ($NO^+$) and 36 also agreed with those predicted from the solution concentrations to within ~15%, whereas the SN/SC ratios estimated from $m/z$ 23 were significantly lower than the predicted values. Based on these experimental results, the applicability of the rTDMS to ambient measurements of sea salt particles is discussed.

## 1 Introduction

Aerosols have large impacts on the Earth's radiation budget by scattering and absorbing solar short-wave radiation (direct effect) and by acting as cloud condensation nuclei (indirect effect) (IPCC 2013). The size and chemical composition of aerosol particles are important for quantitatively estimating the direct and indirect effects of aerosols. Sea salt aerosols

generally make the largest contribution to the budget of natural aerosols in the troposphere (Seinfeld and Pandis, 2006; IPCC, 2013). Significant conversion of sodium chloride (NaCl: SC) to sodium sulfate ($Na_2SO_4$: SS) or sodium nitrate ($NaNO_3$: SN) can take place via the reactions with sulfuric acid ($H_2SO_4$) or nitric acid ($HNO_3$) in polluted air exported from urban areas to coastal regions (Kerminen et al., 1998; Hsu et al., 2007; AzadiAghdam et al., 2019). Biomass burning also makes large contributions to the global budget of soot, organics, and inorganic compounds (Andreae and Merlet, 2001; Akagi et al., 2011; Song et al., 2018). Potassium chloride (KCl: PC) is the dominant form of chloride in biomass burning aerosols (Reid et al., 2005). Potassium may also be present in biological particles emitted from rainforest (Pöhlker et al., 2012). Significant conversion of PC to potassium sulfate ($K_2SO_4$: PS) or potassium nitrate ($KNO_3$: PN) can also take place via the reactions with $H_2SO_4$ or $HNO_3$ when smoke from biomass burning mixes with urban pollution (Li et al., 2003; Zauscher et al., 2015; Schlosser et al., 2017).

Earlier studies investigated the chemical transformation of $Cl^-$ to $SO_4^{2-}$ or $NO_3^-$ in sea salt or biomass burning particles by offline transmission electron microscopy (TEM) and X-ray spectrometry (Miura et al., 1991; Li et al., 2003; Adachi et al., 2015). Single-particle mass spectrometry was also used to qualitatively detect the chemical transformation of $Cl^-$ to $SO_4^{2-}$ and $NO_3^-$ (Hayes et al., 2013). Aerosol particles formed via the chemical transformation processes include both non-refractory and refractory compounds. Note that the definition of "non-refractory" and "refractory" compounds in atmospheric aerosols is rather empirical and depends on the analysis method. Following the definition by Kobayashi et al. (2021), chemical compounds with a bulk thermal desorption temperature lower than ~673 K are referred to as non-refractory compounds, and others are referred to as refractory compounds. As briefly reviewed by Kobayashi et al. (2021), currently available techniques for online quantitative measurements of aerosols, including a particle-into-liquid-sampler coupled with ion chromatography (PILS-IC; Weber et al., 2001) and an Aerodyne aerosol mass spectrometer (AMS; Jayne et al., 2000), are not optimized for separate quantification of refractory and non-refractory aerosols. The PILS-IC can measure total water-soluble ions but is not designed to specify their chemical form.

The primary purpose of Kobayashi et al. (2021) was to develop a new instrument to quantify non-refractory and refractory sulfate aerosols, which was referred to as a refractory aerosol thermal desorption mass spectrometer (rTDMS). The goal of the present study is to develop a new method to quantify particulate sodium and potassium salts (nitrate, chloride, and sulfate) by using the rTDMS. Potential application to sea salt aerosols is also discussed.

## 2 Experimental

### 2.1 Instrument description

The concept and operation procedures of the rTDMS were described in detail by Kobayashi et al. (2021), and thus only the key points are presented here. The configuration is similar to that of the particle trap - laser desorption mass spectrometer (PT-LDMS) (Takegawa et al., 2012). The major difference between the rTDMS and PT-LDMS is the structure of the particle collector and laser power. The main components of the rTDMS include an aerodynamic lens (ADL), a graphite

particle collector, a quadrupole mass spectrometer (QMS) equipped with a cross-beam type electron ionization (EI) source (QMG700, Pfeiffer Vacuum), and a continuous wave focused $CO_2$ laser (wavelength: 10.6 μm, ULR-25, Universal Laser Systems). Aerosol particles are introduced into a vacuum chamber via the ADL and collected on the graphite collector. The structure of the ADL is the same as that used by Miyakawa et al. (2014), which is essentially identical to that presented by Zhang et al. (2004). During particle loadings with the inlet valve open, the graphite collector faces the direction toward the inlet. After particles are collected, the graphite collector turns to the opposite direction so that it faces the ionizer of the QMS. The collected particles are vaporized by the $CO_2$ laser, and evolved gas molecules are detected using the QMS. The blackbody equivalent temperature of the outer surface of the graphite collector was measured using a radiation thermometer (Impac IGA 140, LumaSense Technologies, Inc.). The focused $CO_2$ laser coupled with the graphite collector is the key component of the rTDMS and enables a high desorption temperature (blackbody equivalent temperature of up to 930°C) and a fast increase of the temperature (less than 60 s to reach the maximum temperature from room temperature). The distance between the graphite collector and the ionizer was shortened to 25 mm from 50 mm used by Kobayashi et al. (2021), increasing the sensitivity by a factor of two to three.

The rTDMS was operated with total measurement cycles ranging from 6 to 10 min, including the time for particle collection (2–6 min), laser irradiation and ion detection (2 min), and cooling of the graphite collector (2 min). The graphite collector was automatically rotated to set the position by using an electronic actuator. The sample was irradiated by the $CO_2$ laser for 60 s. The inlet valve was kept closed for 4 min during the laser desorption analysis and cooling. Kobayashi et al. (2021) employed a constant laser power (~20 W) for the detection of non-refractory and refractory sulfate. In this study we tested a two-step laser modulation for better separation of multi-component aerosol particles. Details of the laser power settings are described in Sect. 3.1 and the Supplement.

The QMS was operated in the multiple ion detection (MID) mode to measure ion signals at some selected $m/z$ peaks. The number of selected ion signals was typically limited to less than 10 to obtain better signal-to-noise ratios. We measured all possible $m/z$ signals for each compound to identify the major $m/z$ peaks that can be used for the quantification (see Table S1 in the Supplement).

## 2.2 Laboratory experiments

The experimental apparatus was basically the same as that presented by Kobayashi et al. (2021). The particle generation system included an air compressor, a Collison atomizer (Model 3076, TSI, Inc.), a diffusion dryer (Model 3062, TSI, Inc.), and a differential mobility analyzer (DMA; Model 3080, TSI, Inc.). A condensation particle counter (CPC; Model 3022A, TSI, Inc.) and the rTDMS were connected downstream of the particle generation system. We set the mobility diameter at 200 nm to generate monodisperse particles. The mass loadings of monodisperse aerosol particles were calculated from the CPC data with corrections for multiply-charged particles (Takegawa and Sakurai, 2011) to derive the instrument sensitivity to various types of particles. Particle-free air (zero air: ZA) was introduced into the rTDMS and CPC to correct for blank levels and also for evaluating potential artifacts.

The chemical compounds tested in this study are summarized in Table 1. The main test particles included single-component SN, SC, SS, PN, PC, and PS particles, internally mixed multi-component SN + SC + SS and PN + PC + PS particles with a molar ratio of 1:1:1, and laboratory-generated sea salt particles. Ammonium nitrate ($NH_4NO_3$: AN), ammonium chloride ($NH_4Cl$: AC), and ammonium sulfate (($NH_4)_2SO_4$: AS) particles were also used for evaluating the interference from ammonium salts. The results of the measurements of externally-mixed ammonium salt particles are shown in the Supplement.

Under real-world conditions, interferences from coexisting compounds should be carefully considered. Regarding sea salt aerosols, this can be achieved by using seawater samples for generating calibration particles. We sampled seawater from Tokyo Bay and prepared a diluted solution (by a factor of 100; equivalent to 4.5 mM of NaCl). The diluted seawater was used for generating pure sea salt particles. We also added SN and SS into the diluted seawater to prepare samples for the standard addition method. The tested solutions are summarized in Fig. 1. The seawater experiment aimed to quantify SN and SS in the presence of unidentified compounds in seawater. Estimation of the molar concentrations of various salts in the seawater sample is described in the Supplement. It should be noted that the simple atomization of seawater samples may not reproduce real-world sea salt particles generated by wave breaking-driven bubble bursting (Fuentes et al., 2010). Therefore, the molar fractions of the inorganic compounds in the laboratory-generated particles may be different from those in real-world sea salt particles. Regarding biomass burning aerosols, it is not straightforward to generate combustion particles with reasonable reproducibility. Therefore, we only performed the experiments with authentic standard materials containing potassium.

**Table 1.** Chemical compounds tested in this study [a].

| Compound | Abbreviation | Density (g cm$^{-3}$) | Bulk decomposition temperature (°C) |
|---|---|---|---|
| $NaNO_3$ | SN | 2.26 | 491 |
| NaCl | SC | 2.16 | 800 [b] |
| $Na_2SO_4$ | SS | 2.66 | 850 |
| $KNO_3$ | PN | 2.11 | 526 |
| KCl | PC | 1.98 | 770 [b] |
| $K_2SO_4$ | PS | 2.66 | 907–1395 |
| $NH_4NO_3$ | AN | 1.72 | 142–300 |
| $NH_4Cl$ | AC | 1.53 | 220–360 |
| $(NH_4)_2SO_4$ | AS | 1.77 | 250–500 |

a. The values of density were taken from Nagakura et al. (1998). The bulk decomposition temperatures were taken from Broström et al., 2013; Tagawa, 1987; Halle and Stern, 1980; Halsted, 1970; Nakamura et al., 1980; and Tatykaev et al., 2014. The data for ammonium salts are only shown in the Supplement.

b. These values are the melting points.

**(a) Single component**

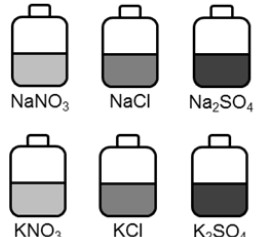

**(b) Multi component**

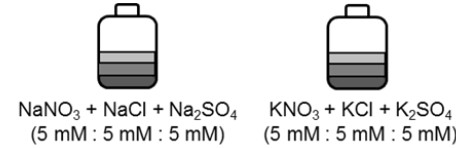

**(c) Seawater**

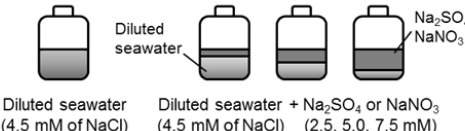

Figure 1. Summary of tested solutions. (a) Single-component solution of NaNO₃ (SN), NaCl (SC), Na₂SO₄ (SS), KNO₃ (PN), KCl (PC), and K₂SO₄ (PS) with a molar concentration of 5 mM. (b) Multi-component solutions of NaNO₃ + NaCl + Na₂SO₄ and KNO₃ + KCl + K₂SO₄ with a molar ratio of 1:1:1 (5 mM : 5 mM : 5 mM). (c) Diluted seawater (by a factor of 100; equivalent to 4.5 mM of NaCl) and diluted seawater with Na₂SO₄ or NaNO₃ with molar concentrations of 2.5, 5.0, and 7.5 mM.

### 2.3 Data analysis

The equations for calculating the mass loadings for each compound $i$, $W_i$, are the same as those presented by Kobayashi et al. (2021) for both the single-component and multi-component experiments. For a single compound $i$, the mass

loadings of the cation or anion of the compound $i$ ($i$ = Na$^+$, K$^+$, NO$_3^-$, Cl$^-$, and SO$_4^{2-}$) introduced into the rTDMS, $W_i$, were calculated as follows:

$$W_i = \frac{\pi}{6} d_m^3 \rho_i f_i N_{\text{CPC}} F t \qquad (1)$$

where $d_m$ is the mobility diameter of particles, $N_{\text{CPC}}$ is the number concentration of monodisperse particles measured by the CPC, $F$ is the sample flow rate, and $t$ is the particle collection time. $\rho_i$ and $f_i$ are the bulk material density and the mass fraction

of the cation or anion, respectively, of the compound $i$. For multi-component particles, we used the following equations:

$$\frac{\pi}{6}d_m^3 = \frac{n}{N_A}\sum_i \frac{a_i M_i}{\rho_i} \qquad (2)$$

$$W = N_{CPC}Ft\frac{n}{N_A}\sum_i f_i a_i M_i \qquad (3)$$

where $n$ is the total number of molecules in each particle, $N_A$ is the Avogadro's number, and $W$ is the mass loadings of multi-component compounds introduced into the rTDMS. $a_i$ and $M_i$ are the molar fraction and the molecular weight, respectively, of the compound $i$ ($\Sigma a_i = 1$). The value of $n$ derived from Eq. (2) is used to calculate the value of $W$ by Eq. (3). The effects of hydrated water should be properly considered for the calculations. The relative humidity (RH) downstream of the diffusion dryer was ~5%. Previous studies showed that the efflorescence RHs (ERHs) were 43, 56, 59, and 60% for SC, SS, PC, and PS particles, respectively (Seinfeld and Pandis, 2006; Freney et al., 2009). We assumed that these compounds were effloresced to form solid crystals before reaching the DMA. Previous studies also reported that nitrate particles did not exhibit clear ERH features (e.g., Freney et al., 2009; Lee and Hsu, 2000). We assumed that SN and PN particles lost a majority of water molecules in order to calculate the mass of particles.

Laboratory experiments for Aerodyne AMSs showed that the collection efficiencies of non-refractory particles are largely controlled by particle bounce on the vaporizer (Matthew et al., 2008; Robinson et al., 2017; Saleh et al., 2017). Matthew et al. (2008) showed that laboratory-generated solid crystalline AS particles exhibited a particle collection efficiency of ~20%, which appears to be near the lower bound of the collection efficiencies for various types of particles. The particle collection efficiencies for the rTDMS would be mostly controlled by particle bounce effects as long as the temperature induced by the laser heating exceeds the vaporization temperature of the particle compounds.

Kobayashi et al. (2021) showed that the collection efficiency for solid AS, PS, and SS particles was ~70%. We did not measure the collection efficiencies for various types of particles tested in this study. Following the experimental results for the Aerodyne AMSs, we assume that the collection efficiencies for various types of particles were comparable to or higher than that for the laboratory-generated solid crystalline AS particles (~70%) in the rTDMS. For SC and PC particles, the collection efficiencies would be ~70% because they were likely in the form of solid crystals. For SN and PN particles, the particle collection efficiencies would be in the range of ~70–100% because they might not be in the form of solid crystals.

Temporal profiles of ion signals at selected $m/z$ values were integrated over the laser irradiation time to obtain the integrated ion signals for each compound $i$, $Q_{m/z,i}$. The major $m/z$ values for each compound are listed in Table 2. Variations in $Q_{m/z,i}$ due to drifts in the detector sensitivity were corrected by using $m/z$ 14 ($N^+$) signals. The sensitivity of the rTDMS at $m/z$ to compound $i$, $S_{m/z,i}$, is defined as the ratio of $Q_{m/z,i}$ to $W_i$. Kobayashi et al. (2021) suggested that the ion signals for multi-component sulfate particles could be approximated as the linear combination of ion signals originating from single-component sulfate particles based on mass closure tests. We did not perform detailed mass closure tests in the current study. Alternatively, we compared the SN/SC and SS/SC ratios estimated from the QMS ion signals with those predicted from the ionic concentrations in the solutions.

**Table 2.** Major $m/z$ signals for each compound.

| | $m/z$ |
|---|---|
| SN | 23 ($Na^+$), 30 ($NO^+$) |
| SC | 23 ($Na^+$), 36 ($H^{35}Cl^+$) |
| SS | 23 ($Na^+$), 48 ($SO^+$), 64 ($SO_2^+$) |
| PN | 30 ($NO^+$), 39 ($K^+$, $C_3H_3^+$) |
| PC | 36 ($H^{35}Cl^+$), 39 ($K^+$, $C_3H_3^+$) |
| PS | 39 ($K^+$, $C_3H_3^+$), 48 ($SO^+$), 64 ($SO_2^+$) |

## 3 Results

### 3.1 Single-component particles

Figure 2 shows temporal profiles of ion signals at $m/z$ 23 ($Na^+$), 30 ($NO^+$), 36 ($H^{35}Cl^+$), 39 ($K^+$, $C_3H_3^+$), 48 ($SO^+$), and 64 ($SO_2^+$) originating from monodisperse SN, SC, SS, PN, PC, and PS particles with mobility diameters of 200 nm. As

described earlier, we used a two-step laser modulation for better separation of multi-component aerosol particles. We tested various combinations of laser power settings (power and duration time), as described in the Supplement. The power setting and duration of the first step were 7.5 W and 40 s, and those of the second step were 20 W and 20 s. Note that these laser power values were not measured ones but estimated from the product specifications.

Ion signals at $m/z$ 30 originating from SN particles exhibited a distinct peak at the elapsed time of 7–12 s, whereas

those at $m/z$ 23 exhibited bimodal broad peaks at 7–12 s and 15–50 s (Fig. 2a). The ion signals at $m/z$ 46 ($NO_2^+$) were negligibly small as compared to those at $m/z$ 30, indicating that nitrate was mainly decomposed to NO. Ion signals at $m/z$ 23 and 36 originating from SC particles exhibited a single peak starting at ~8 s with significant tailing up to ~30 s (Fig. 2b). Ion signals at $m/z$ 58 ($Na^{35}Cl^+$) and 81 ($Na_2^{35}Cl^+$) were negligibly small as compared to those at $m/z$ 23 or 36, indicating that the contributions of the intact form of NaCl or its clusters were not significant. Ion signals at $m/z$ 23, 48, and 64 from SS particles

showed a distinct single peak at 40–52 s (Fig. 2c). Following the discussion by Kobayashi et al. (2021), the fragment ratio of $m/z$ 48 to 64 suggests that the major thermal decomposition product from SS particles was $SO_2$ or $SO_3$.

Ion signals at $m/z$ 30 originating from PN particles exhibited a distinct peak at 6–13 s, whereas those at $m/z$ 39 exhibited bimodal broad peaks at 6–13 s and 40–50 s (Fig. 2d). Similarly to SN, ion signals at $m/z$ 46 were negligibly small as compared to those at $m/z$ 30. The ion signals at $m/z$ 39 from PN particles did not reach the background level after the second

peak, suggesting that PN particles were not fully vaporized at the current laser power settings. This may lead to underestimation of the sensitivity at $m/z$ 39 for PN particles. Ion signals at $m/z$ 36 and 39 originating from PC particles exhibited a single peak starting at ~8 s with significant tailing up to ~30 s (Fig. 2e). Similarly to SC, ion signals at $m/z$ 74 ($K^{35}Cl^+$) were negligibly

small as compared to those at $m/z$ 36 or 39. A small increase in the ion signals at $m/z$ 39 from PC particles was observed after 40 s, indicating that PC particles were not fully vaporized by the first laser power setting (7.5 W for 40 s). We estimated the effect of the small peak to be ~20% of that of the main peak by comparing the ion signals after 40 s with those before 40 s. Ion signals at $m/z$ 39, 48, and 64 from PS particles showed small increases at ~30 s followed by a distinct peak at 40–47 s (Fig. 2f). The ion signals at $m/z$ 39 from PS particles also exhibited a peak at ~6–13 s, which was probably due to artifacts ($C_3H_3^+$ from organic compounds adsorbed on the graphite collector). Similarly to SS, the fragment ratio of $m/z$ 48 to 64 suggests that the major thermal decomposition product from PS particles was $SO_2$ or $SO_3$. The timing of the ion signals was qualitatively consistent with the order of the bulk thermal decomposition temperature (Table 1).

Figure 3 shows scatterplots of integrated ion signals ($Q_{m/z,i}$) versus the mass loadings of the corresponding parent compounds ($W_i$) for single-component SN, SC, SS, PN, PC, and PS particles. Data obtained on different days are collectively plotted. The time window for the ion signal integration was set at 5–15 s ($m/z$ 30) and 5–50 s ($m/z$ 23) for SN, 5–15 s ($m/z$ 30) and 5–55 s ($m/z$ 39) for PN, 5–40 s for SC and PC, 40–55 s for SS, and 30–50 s for PS. There was no systematic difference in the $Q_{m/z,i}$–$W_i$ relationship between the data obtained on different days, and the correlation coefficients were generally high ($r^2 > 0.8$) for all the compounds tested, indicating good linearity and reproducibility in detecting these compounds. The scatter in the $m/z$ 39 signals at mass loadings of zero (i.e., ZA measurements) might be due to interference from organic compounds ($C_3H_3^+$). The integrated ion signals at $m/z$ 64 showed good correlation with those at $m/z$ 48 for SS and PS ($r^2 > 0.99$; not shown in Fig. 3).

The difference in the regression slopes at a specific $m/z$ between different compounds can be interpreted as the variability in the sensitivity with respect to the difference in the chemical form. The regression slope at $m/z$ 30 was 49 and 60 (average of 55) pC per ng of $NO_3^-$ for SN and PN, respectively. The regression slope at $m/z$ 36 was 11 and 17 (average of 14) pC per ng of $Cl^-$ for SC and PC, respectively. As mentioned above, the integrated ion signals at $m/z$ 64 showed good correlation with those at $m/z$ 48 for SS and PS, and we used $m/z$ 48 as a representative ion signal for these sulfate compounds. The regression slope at $m/z$ 48 was 5.5 and 4.2 (average of 4.9) pC per ng of $SO_4^{2-}$ for SS and PS, respectively. These results suggest that the variability in the sensitivity was less than ~20% for $NO_3^-$, $Cl^-$, and $SO_4^{2-}$. By contrast, the regression slope at $m/z$ 23 was 24, 16, and 8.6 (average of 16) pC per ng of Na for SN, SC, and SS, respectively, and the regression slope at $m/z$ 39 was 30, 21, and 6.1 (average of 19) pC per ng of K for PN, PC, and PS, respectively. These results suggest that the variability in the sensitivity was considerably large for Na and K (more than ~50%). Furthermore, the sensitivities at $m/z$ 30 for nitrate particles were much larger than those for the other $m/z$ peaks (Figs. 2 and 3). The difference in the sensitivities cannot be explained by the difference in the electron ionization cross sections. We have not identified the mechanisms that caused the variability in the sensitivity values. The difference in the divergence angle of evolved gas molecules after the thermal desorption might be a possible mechanism (Uchida et al., 2019; Ide et al., 2019).

## 3.2 Multi-component particles

Figure 4 shows the temporal evolution of ion signals at $m/z$ 23, 30, 36, and 39 originating from internally mixed, multi-component sodium and potassium salt particles with a molar ratio of 1:1:1. The multi-component ion signals at $m/z$ 23 and 39 exhibited trimodal peaks, with the first and second ones partially overlapped. Based on the comparison with the single-component experiment data, the first, second, and third peaks can be attributed to nitrate (SN, PN), chloride (SC, PC), and sulfate (SS, PS), respectively. These results indicate that the temporal profiles of ion signals at $m/z$ 23 (or 39) originating from internally mixed SC + SS + SN (or PC + PS + PN) particles were characterized by three sequential peaks associated with the evolution of the desorption temperature. Note that the third peak at $m/z$ 23 and 39 might be affected by the tails of the ion signals originating from SC and PC particles, respectively (especially for PC). We have not investigated the effects of the enhanced background signals on the quantification of sodium and potassium in the current study.

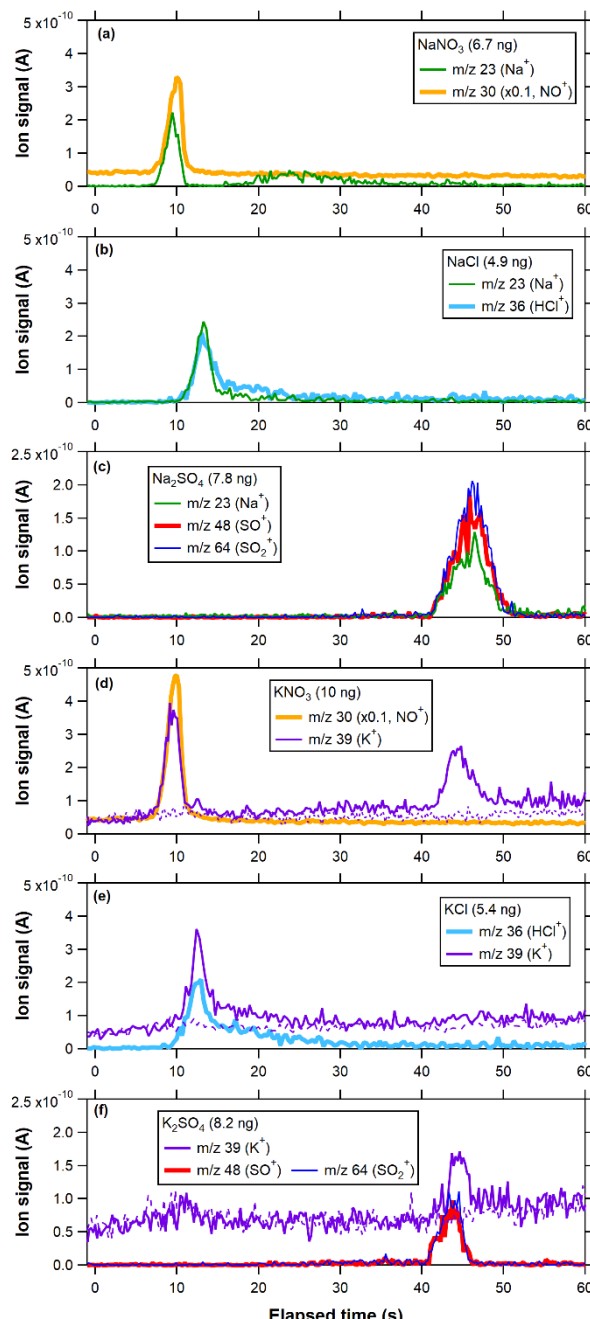

**Figure 2.** Temporal profiles of ion signals originating from single-component (a) SN, (b) SC, (c) SS, (d) PN, (e) PC, and (f) PS particles as a function of elapsed time since turning the laser on. The ion signals at $m/z$ 30 were scaled by a factor of 0.1 for clarity of the presentation. Zero air signals at $m/z$ 39 are also plotted in (d)–(f). Zero signals at $m/z$ 23, 30, 36, 48, and 64 are not displayed because the signal levels were comparable to the baselines (i.e., ion signals before the laser irradiation). Particle mass loadings (in ng) are shown in the legend. The elapsed time may contain an error of ~1 s because the laser operation was performed manually.

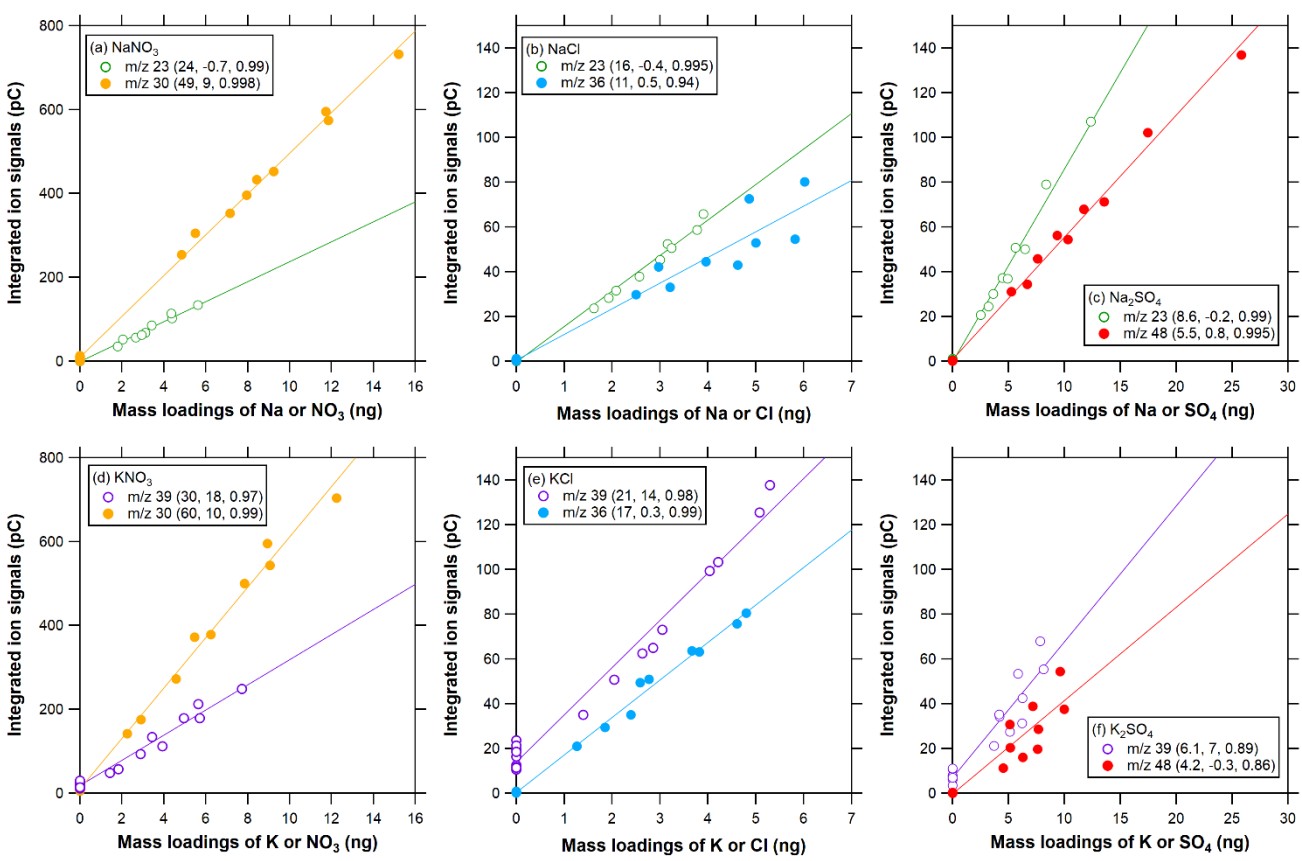

**Figure 3.** Scatterplots of integrated ion signals versus mass loadings (Na for *m/z* 23; K for *m/z* 39; NO$_3$ for *m/z* 30; Cl for *m/z* 36; SO$_4$ for *m/z* 48) for (a) SN, (b) SC, (c) SS, (d) PN, (e) PC, and (f) PS particles. The three values in the parentheses in the legend represent the slope, intercept, and r$^2$, respectively, for the linear regression.

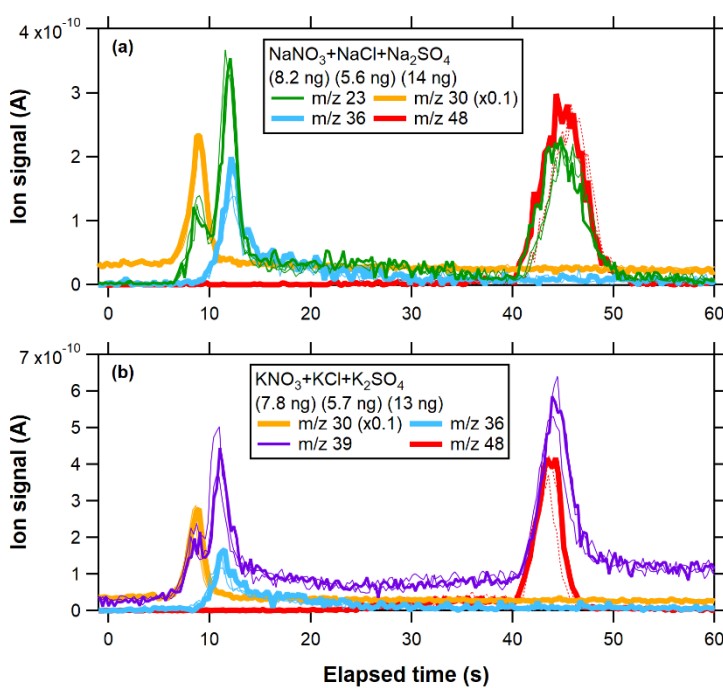

**Figure 4.** Temporal profiles of ion signals originating from an internal mixture of (a) SN + SC + SS and (b) PN + PC + PS particles. The ion signals at $m/z$ 30 were scaled by a factor of 0.1 for clarity of the presentation. The thin and dashed lines represent repeated measurement data normalized to the mass loadings of the main data (the thick lines).

### 3.3 Particles generated from seawater samples

Figures 5a–c show the temporal evolution of ion signals at $m/z$ 23, 30, 36, and 48 originating from diluted seawater and "seawater + SN (or SS)" particles. The ion signals at $m/z$ 23 and 36 originating from diluted seawater particles exhibited temporal evolution shapes similar to those of the single-component SC particles. Ion signals at $m/z$ 23 originating from seawater + SN particles exhibited partially overlapped bimodal peaks.

Let MX be a salt formed from $M^+$ and $X^-$ ions. We define the equivalent molar concentration of MX in the seawater sample as the molar concentration of $M^+$ (or $X^-$) that would form MX by complete dehydration (see the Supplement for details). Figure 6 shows scatterplots of the molar ratios of SN (or SS) to SC in collected particles estimated from ion signals versus the equivalent molar ratios of SN (or SS) to SC in the solutions. The data points for the authentic multi-component particles described in Sect. 3.2 are also plotted for comparison. The molar ratios of SN (or SS) to SC in the collected particles were calculated using the integrating ion signals at $m/z$ 30 (or 48) and 36. We also used ion signals at $m/z$ 23 to estimate the molar ratios of SN (or SS) to SC in collected particles. We employed a multi-peak fitting method for both the single- and multi-component sodium salt particles to estimate the contributions of SN and SC to the measured ion signals at $m/z$ 23 (see the Supplement). The linear regression slopes for the seawater + SN particles were found to be 0.48 (by $m/z$ 23) and 0.85 (by $m/z$

30 and 36), and those for the seawater + SS particles were found to be 0.98 (by *m/z* 23) and 1.1 (by *m/z* 36 and 48). The data for the seawater + SN (or SS) samples showed a good linearity ($r^2 \geq 0.98$) and agreed well with the data points for the authentic multi-component particles, suggesting that interferences of unidentified compounds in seawater were small. The SS/SC ratios estimated from the ion signals at *m/z* 23, 36, and 48 agreed well with those predicted from the solution concentrations to within ~10%. The SN/SC ratios estimated from the ion signals at *m/z* 30 and 36 also agreed with those predicted from the solution concentrations to within ~15%, whereas the SN/SC ratios estimated from the ion signals at *m/z* 23 were significantly lower (~52%) than the predicted values. Possible causes for the difference will be discussed in Sect. 4.3.

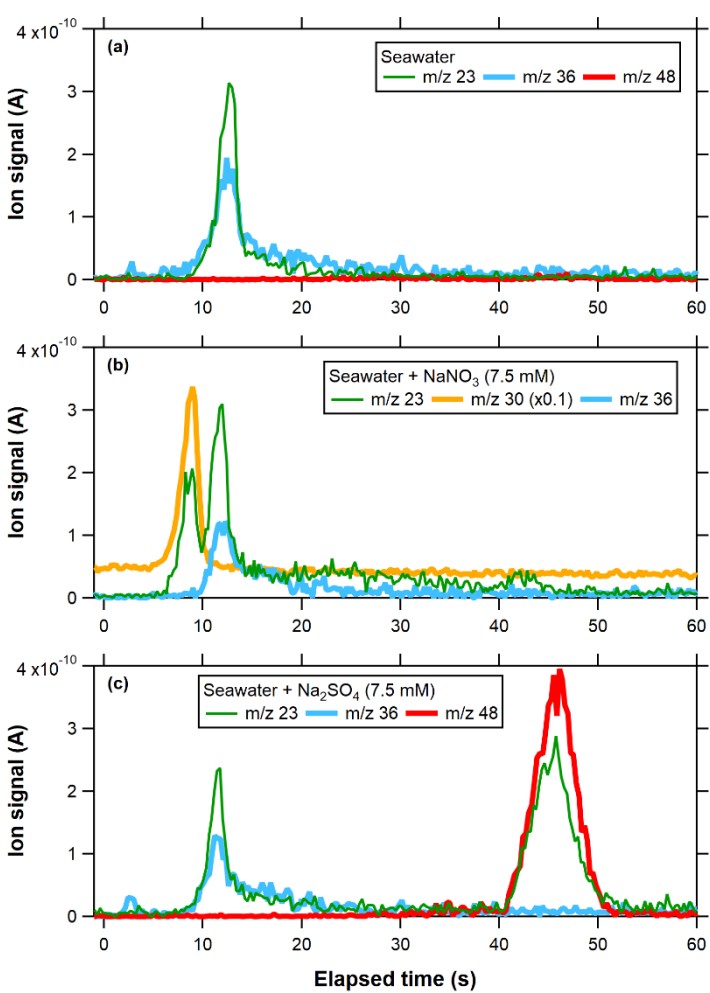

**Figure 5.** Temporal profiles of ion signals originating from (a) seawater (equivalent to 4.5 mM of SC), (b) seawater + SN (7.5 mM), and (c) seawater + SS (7.5 mM) particles.

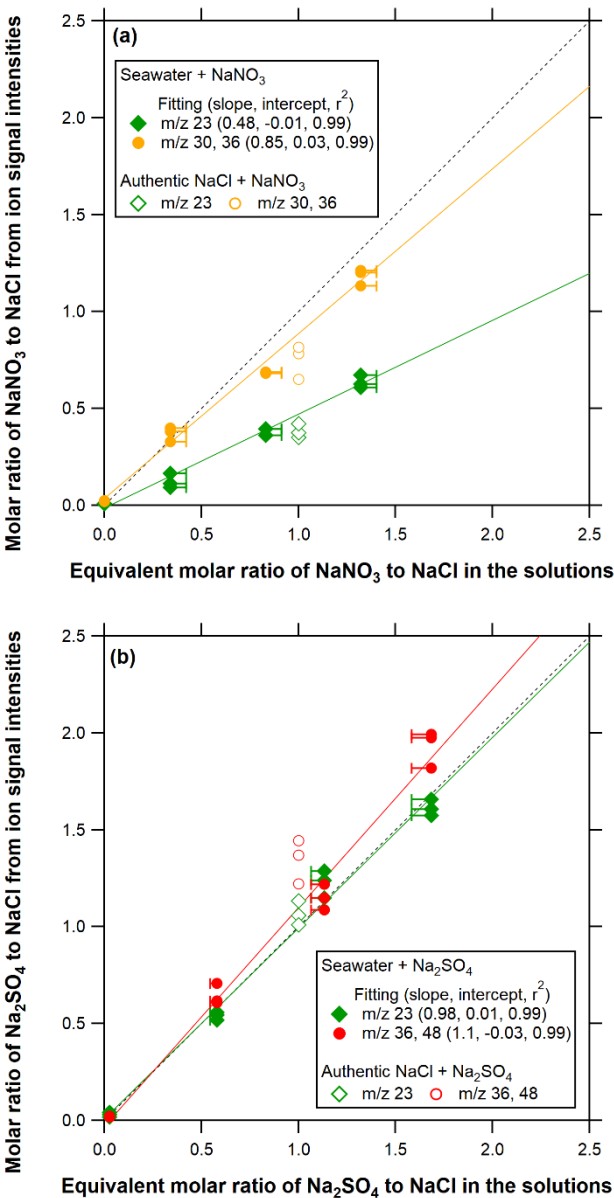

**Figure 6.** (a) Scatterplots of the molar ratios of SN to SC derived from ion signals versus the equivalent molar ratios of SN to SC in the solutions. The slope, intercept, and $r^2$ values for the linear regression are indicated in the legend. Solid and open symbols represent the data from seawater + SN and SC + SN particles, respectively. The error bars for the x-axis represent the uncertainties in estimating the equivalent molar concentrations of the compounds (see the Supplement for details). (b) Same as (a) but for the molar ratios of SS to SC.

## 4 Discussion

### 4.1 Thermal decomposition mechanisms

The thermal decomposition mechanisms for SS and PS were discussed by Kobayashi et al. (2021). Here we discuss possible thermal decomposition processes of nitrate and chloride. Our experimental data indicate that the thermal decomposition of SN particles yielded gas-phase Na and NO, and that of PN particles yielded gas-phase K and NO. However, the temporal evolution of the ion signals at $m/z$ 23 and 39 suggests that the thermal decomposition processes of SN and PN particles were not represented by single-step reactions. Tagawa (1987) proposed the following reactions for thermal

decomposition of bulk SN in dry air:

$$NaNO_3 \rightarrow NaNO_2 + 1/2\ O_2\ (\sim491\text{--}750°C)$$
$$NaNO_2 \rightarrow 1/2\ Na_2O_2 + NO\ (\sim750\text{--}850°C)$$
$$Na_2O_2 \rightarrow Na_2O + 1/2\ O_2\ (> 850°C)$$

and those for PN in dry air:

$$KNO_3 \rightarrow KNO_2 + 1/2\ O_2\ (\sim526\text{--}750°C)$$
$$KNO_2 \rightarrow 1/2\ K_2O_2 + NO\ (\sim750\text{--}900°C)$$
$$K_2O_2 \rightarrow K_2O + 1/2\ O_2\ (> 900°C)$$

We speculate that $Na_2O$ and $K_2O$ underwent further thermal decomposition reactions to yield gas-phase Na and K in the rTDMS. The observed bimodal peaks at $m/z$ 23 (or 39) originating from SN (or PN) particles might be attributed to the

sequentially occurring thermal decomposition of $NaNO_3$ (or $KNO_3$) and $NaO_x$ (or $KO_x$).

    The thermal decomposition processes of chloride would be simpler than those of nitrate and sulfate. We did not observe significant ion signals at $m/z$ 58 ($Na^{35}Cl^+$) and 74 ($K^{35}Cl^+$). Therefore, SC and PC particles were likely decomposed to metallic and halogen atoms:

$$NaCl \rightarrow Na\ (g) + Cl\ (g)$$
$$KCl \rightarrow K\ (g) + Cl\ (g)$$

The sensitivity of ion signals at $m/z$ 36 was larger than those at $m/z$ 35 by a factor of $\sim$3. This suggests that Cl (g) evolved from SC or PC particles mostly reacted with $H_2O$ in background air or on the graphite surface to form HCl (g) (Drewnick et al., 2015; Tobler et al., 2020).

    As described in Sect. 2.1, we used the two-step laser modulation to separately quantify the aerosol compounds tested

in this study. While this method successfully separated SS from SN and SC, the ion signals for SN and SC showed temporal overlapping. This is somewhat unexpected considering that the difference in the bulk thermal desorption temperature between SN and SC is larger than that between SC and SS (Table 1). The cause of this discrepancy is currently unknown.

## 4.2 Uncertainties in the quantification of single-component particles

The limit of detection (LOD) was estimated as the equivalent concentration at three times the standard deviation (3σ) of the integrated ion signals for repeated ZA measurements (20 samples for each compound). The equivalent concentration is calculated as the ratio of the 3σ value to the product of the sensitivity at a certain $m/z$ and sample air volume. The LOD values estimated by this method are listed in Table 3.

The systematic errors due to the particle collection efficiencies were already described in Sect. 2.3. Other systematic errors may originate from the accuracy of the CPC detection efficiency, the DMA sizing, and the flow rate calibrations. These factors are not considered here because they are not specific to the detection of sodium and potassium salt particles by the rTDMS. The systematic errors might also originate from the effective density of particles, which was discussed in our previous study (Kobayashi et al., 2021). This factor is not considered because it was difficult to quantify with the current experimental apparatus.

**Table 3.** Limit of detection (LOD; µg m$^{-3}$) for the mass concentrations of SN, SC, SS, PN, PC, and PS particles. The values in parentheses denote the mass loadings (ng).

|     | Cation signal ($m/z$ 23, 39) | Anion signal ($m/z$ 30, 36, 48) |
| --- | --- | --- |
| SN | 0.06 (0.04 ng) | 0.5 (0.3 ng) |
| SC | 0.1 (0.08 ng) | 0.1 (0.07 ng) |
| SS | 0.1 (0.07 ng) | 0.08 (0.06 ng) |
| PN | 0.4 (0.2 ng) | 0.4 (0.3 ng) |
| PC | 0.5 (0.3 ng) | 0.07 (0.05 ng) |
| PS | 0.9 (0.6 ng) | 0.1 (0.09 ng) |

The LODs were estimated as the equivalent concentration at three times the standard deviation (3σ) of the integrated ion signals for repeated ZA measurements with a total measurement cycle of 10 min (particle collection time of 6 min). The 10-min measurement cycle was chosen because it was used for ambient test measurements.

## 4.3 Uncertainties in the quantification of multi-component sodium salt particles

The systematic errors in quantifying the SN/SC and SS/SC ratios (Fig. 6) are discussed. The collection efficiency of SC and SS particles could be approximated as ~70% because they were likely in the form of solid crystals (see Sect. 2.3). Assuming that the seawater (mainly SC) + SS and SC + SS particles were in the form of solid crystals, the collection efficiency would also be approximated as ~70%. Therefore, the uncertainty in the SS/SC ratios due to the particle collection efficiencies was probably small. For the seawater + SN and SC + SN particles, however, the interpretation of the collection efficiency is rather complicated. The collection efficiency of the SN particles would be in the range of ~70–100% because they might not be in the form of solid crystals (see Sect. 2.3). Ishizaka et al. (2019) showed that the ERH of equimolar mixed SC and SN

particles was ~33%, suggesting that the SC + SN particles (either authentic materials or seawater) were in the solid phase when introduced into the rTDMS. Assuming that the collection efficiency of the SC + SN particles was approximated as ~70%, the SN/SC ratios could be underestimated by up to ~30% when using the sensitivity for the SN particles.

The deviation of the SN/SC ratios derived from the $m/z$ 23 signals (~52%) cannot be explained by the systematic errors due to the collection efficiency. We compared the temporal profiles of ion signals at $m/z$ 23 from the single-component SN and SC particles and multi-component SC + SN particles (see the Supplement for details). The ion signals at $m/z$ 23 that could be attributed to SN in the multi-component particles (i.e., the first peak) were smaller than those expected from the single-component SN particles. By contrast, the ion signals at $m/z$ 23 that could be attributed to SC in the multi-component particles (i.e., the second peak) were larger than those from the single-component SC particles. These results suggest that there might be significant matrix effects in the quantification of SN and SC particles when using ion signals at $m/z$ 23 (i.e., the sensitivities depend on coexisting compounds). We have not investigated the mechanisms responsible for the matrix effects. Therefore, ion signals at $m/z$ 23 should be used only for qualitative identification of SN and SC particles.

## 5 Conclusions and Future Outlook

The quantification of sodium and potassium salt (nitrate, chloride, sulfate) particles by the rTDMS was investigated in the laboratory. We generated single-component sodium or potassium salt particles (SN, SC, SS, PN, PC, and PS) and multi-component SN + SC + SS or PN + PC + PS particles. We also generated particles from real seawater samples. The major conclusions are summarized below.

1. Major ion signals originating from single-component sodium or potassium salt particles were clearly detected associated with the increase in the desorption temperature by laser heating. A good linearity ($r^2 > 0.8$) and reproducibility were found between the major ion signals and mass loadings.

2. Temporal profiles of ion signals at $m/z$ 23 (or 39) originating from multi-component sodium (or potassium) salt particles were characterized by three sequential peaks associated with the evolution of the desorption temperature. The first, second, and third peaks were attributed to SN (or PN), SC (or PC), and SS (or PS), respectively.

3. Particles generated from seawater samples were used to test the potential interference or matrix effects in the quantification of SN and SS under real-world conditions. The SS/SC ratios estimated from the ion signals at $m/z$ 23, 36, and 48 agreed well with those predicted from the solution concentrations to within ~10%. The SN/SC ratios estimated from the ion signals at $m/z$ 30 and 36 also agreed with those predicted from the solution concentrations to within ~15%, whereas the SN/SC ratios estimated from $m/z$ 23 were significantly lower than the predicted values. Because the ion signals at $m/z$ 23 originating from the SC and SN particles might be influenced by matrix effects (i.e., the sensitivities depend on coexisting compounds), they should be used only for qualitative identification of SC and SN particles.

The laboratory experiments demonstrated the proof of concept under well-controlled conditions. In the current study we did not perform detailed mass closure tests, and it has not been fully tested whether the ion signals for multi-component particles could be approximated as the linear combination of ion signals originating from single-component particles. The uncertainty

in the particle collection efficiency is one of the key issues, and thus the variability in the particle collection efficiencies due to differences in the chemical composition and mixing states should be minimized. Kobayashi et al. (2021) showed that the collection efficiencies for laboratory-generated solid sulfate particles were ~70%. Ambient particles might be collected with even higher efficiencies if they are in the liquid phase or internally mixed with "sticky" compounds, leading to systematic uncertainties of up to ~30%. Increasing the collection efficiencies for solid particles by improving the geometry of the particle collector may reduce the uncertainties.

Furthermore, modifications of the inlet system and vacuum system are needed to deploy the rTDMS for ambient measurement of sea salt particles. The structure of the current ADL is the same as that used by Miyakawa et al. (2014), which was optimized for efficiently transmitting submicron particles. An alternative ADL for transmitting supermicron particles is desired for measuring ambient sea salt aerosols. Because supermicron ADLs generally require higher operating pressures (Schreiner et al., 1999; Williams et al., 2013), a better differential pumping system may be needed to maintain sufficient vacuum levels for the ion detection.

**Author contribution**

YK and NT designed the research; YK performed the laboratory experiments and data analysis; YK and NT wrote the paper.

**Data availability**

The data used in this study are provided as a supplementary material and will be available at a data repository.

**Competing interests**

The authors declare no conflict of interest.

**Acknowledgments**

This study was funded by Grants-in-Aid for Scientific Research of the Japan Society for the Promotion of Science (JSPS) (16H05620, 17H01862, 20H04310).

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
