# Peer review of "A new method to quantify particulate sodium and potassium salts (nitrate, chloride, and sulfate) by thermal desorption aerosol mass spectrometry"

_Atmospheric Measurement Techniques, 2021_

## Author Comment (AC1)

**Response to Referee 1**

In their manuscript "A new method to quantify particulate sodium and potassium salts (nitrate, chloride, and sulfate) by thermal desorption aerosol mass spectrometry", Kobayashi and Takegawa present measurements of sodium and potassium sulfates, nitrates, and chlorides with their 'refractory aerosol thermal desorption mass spectrometer' (rTDMS). The rTDMS collects particles on a graphite collector and vaporizes them using a focused CO2 laser before electron ionization and quadrupole MS analysis. This instrument was presented in a recent publication by the authors as well as the analysis of various sulfate salts.

In this manuscript, the sulfate salts analysis is presented again, however in a different setting together with chloride and nitrate salt measurements, focusing on the analysis of sodium and potassium salts. The authors present vaporization time series of single- and multi-component samples, analysis of linearity of the analysis, and detection limits for the various components.

While this presentation of measurement capabilities of the rTDMS in little slices (first various sulfate salts, now various sodium and potassium salts – including the sulfates) seems to be an attempt to improve the number of publications on the measurement capabilities of the instrument, they deliver a thorough characterization of the rTDMS capabilities to separate and quantify these kinds of salts with a reasonable attempt to explain the observed features, presented in a clear way. Since the subject of the manuscript fits well into the scope of AMT and since quantitative (semi-)online aerosol mass spectrometry of sodium and potassium salts is still not established in the aerosol community, I recommend publishing this manuscript after the following rather minor issues have been reasonably addressed.

We would like to thank the referee very much for giving us valuable comments and suggestions. We have revised the manuscript to address those comments and also made other corrections to improve the clarity of the presentation. The line numbers for this response letter are based on the manuscript with track changes.

(1) P1L10-11: "refractory" is defined as material that keeps its structural properties at very high temperatures. Examples are oxides or carbides of metals like aluminum or magnesium. In this manuscript, materials are analyzed with the rTDMS that have bulk decomposition temperatures from 142 up to  $850^{\circ}$ C - with the exception of K2SO4 - and at temperatures of the graphite collector up to 930°C. These are temperatures of which most are within the accessible range of e.g. the vaporizer of the Aerodyne AMS (typical 550-600°C, 800°C can be reached), an instrument which claims to measure "non-refractory" aerosol components. I wonder whether the name "refractory TDMS" is adequate for an instrument with these features; most of the really refractory materials could probably not be measured with this instrument.

The definition of "non-refractory" and "refractory" compounds in atmospheric aerosols is rather empirical and depends on the analysis method. The temperature of the graphite collector are mostly within the vaporizer temperature of an Aerodyne AMS (~600°C), as the referee pointed out. However, the temperature of particles that hit the vaporizer of an Aerodyne AMS may not reach the vaporizer temperature because of the particle bounce and latent heat effects (Saleh et al., 2017). To our understanding, non-refractory sulfate, nitrate, and chloride aerosols measured by Aerodyne AMSs are not strictly defined (Drewnick et al., 2015) and nearly equivalent with AS, AN, and AC in most cases.

The terminology "refractory" does not have a strict definition in the rTDMS. Following the definition by Kobayashi et al. (2021), chemical compounds with a bulk thermal desorption temperature lower than ~673 K are referred to as non-refractory compounds, and the others are referred to as refractory compounds. Although the rTDMS may not comprehensively measure refractory aerosols, we consider that the terminology "refractory" is appropriate to represent the general characteristics of our instrument.

P.2, L46-50

(2) P1L29-P2L36: This text largely repeats the information from the lines above it.We have removed the sentences "Aerosol particles emitted from sea spray ... leads to the displacement of chloride relative to sodium (or potassium)" in Section 1.

P1, L30 - P2, L33

(3) P2L42-43: The PILS-IC measures ion concentrations after dissolving the soluble aerosol components in water. It does not care about whether the material is refractory or not, just whether it is soluble or not. All the salts presented in this study could be measured with the PILS-IC.

The PILS-IC can measure all the salts presented in this study, as the referee pointed out. However, the PILS-IC measures total water-soluble ions and is not designed to specify their chemical form. We have added this point in Section 1.

**P2, L53-54**

(4) P4L100: It is unclear to me what the benefit of repeating the information from the text within this figure is. The concentration information could easily be added to the respective information in the text and then the figure could be omitted.

We used multi-component solutions with various mixing ratios and also diluted seawater with the addition of authentic standards. Although the figure does not add further information, it would be helpful to visually understand the combination of the solutions. We would like to keep Fig. 1 as it is. (5) P5L120-121: Instead of presenting the method how mass loadings were calculated, the authors refer to their recent paper (Kobayashi et al., AS&T 2021). For some readers it is quite unfortunate that this paper is not openly accessible, and therefore the open access benefit of AMT is somewhat limited for readers without AS&T access.

We have added Equation (1), (2), and (3) at Section 2.3 to clarify the method for calculating the mass loadings.

**P6, L135-146**

(6) P5L127: The particle collection efficiency discussion in this manuscript is a weak point of the whole presentation. From other particle-collecting devices it is known that particle collection efficiency can vary strongly. Here, some assumptions about collection efficiency were made, however, no measurements were presented which provide a basis for these assumptions. At the end, particle collection efficiency differences between single component particles which might be used for calibration and real-world particles with different components and under different RH conditions could result in much larger errors than the 15-30% uncertainty presented here.

We have expanded the explanation of the collection efficiency in Section 2.3. Laboratory experiments for Aerodyne AMSs showed that the collection efficiencies of non-refractory particles are largely controlled by particle bounce on the vaporizer (Matthew et al., 2008; Robinson et al., 2017; Saleh et al., 2017). Matthew et al. (2008) showed that laboratory-generated solid crystalline AS particles exhibited a particle collection efficiency of ~20%, which appears to be near the lower bound of the collection efficiencies for various types of particles. The particle collection efficiencies for the rTDMS would be mostly controlled by particle bounce effects as long as the temperature induced by the laser heating exceeds the vaporization temperature of the particle compounds.

Kobayashi et al. (2021) showed that the collection efficiency for solid AS, SS, PS, and MgSO4 (MS) particles was approximated as 70%. The variability in the collection efficiencies was relatively small ( $<\sim$ 7%). Based on the experimental results for the Aerodyne AMS, we consider that the collection efficiencies for various types of particles were comparable to or higher than that for the laboratory-generated solid crystalline AS particles ( $\sim$ 70%) in the rTDMS. For SC and PC particles, the collection efficiencies would be  $\sim$ 70% because they were likely in the form of solid crystals. For SN and PN particles, the particle collection efficiencies would be in the range of  $\sim$ 70–100% because they might not be in the form of solid crystals.

**P6, L154 - P7, L166**

(7) P6L140: Some of the ions of Table 2 require some more information. For SS m/z 23 (Na+) and 48 (SO+) are listed in the table. According to Figure 2, the SO2+ signal (m/z 64) is larger

than the m/z 48 signal. Why is it not included in the table? Furthermore, for PN, PC, and PS the  $C_3H_3^+$  ion is listed in the table. What is the origin of this ion?

We have added ion signals at m/z 64 for SS and PS in Table 2. We have also added ion signals at m/z 64 for PS in Figure 2. We consider that the origin of C3H3+ ions (m/z 39) was contamination of organic compounds on the graphite collector.

P7, Table 2; P10, Fig. 2; P8, L210-211

(8) P6L158: For m/z 39 and 48 from PS small increases at  $\sim$ 30s were observed. For m/z 39 also a peak at  $\sim$ 7-15 s can be seen in Figure 2(f). This is not mentioned or explained.

We have added the information on the artifacts in Section 3.1. The small increases in the m/z 39 and 48 signals from PS particles at ~30 s indicates the onset of the thermal decomposition of PS particles. The increases in the m/z 39 signals at ~6–13 s was probably due to artifacts, as mentioned in the answer to the previous question.

**P8, L210-211**

(9) P7L161-167: How was the background signal, i.e. the signal outside the peak integration area, accounted for? For some of the m/z signals in Figure 2 it does not return to zero after the peak, how is this handled?

We have added the description about insufficient vaporization of PN and PC particles in Section 3.1. The ion signals at m/z 39 from PN particles did not reach the background level after the second peak, suggesting that PN particles were not fully vaporized by the current laser power settings. This may lead to underestimation of the sensitivity at m/z 39 for PN particles. Furthermore, a small increase in the ion signals at m/z 39 from PC particles was observed after 40 s, indicating that PC particles were not fully vaporized by the first laser power setting (7.5 W for 40 s). We estimated the effect of the small peak to be ~20% of that of the main peak by comparing the ion signals after 40 s with those before 40 s.

P8, L201-208

(10) P7L164: Please define "Qi" and "Wi".

The definition of  $Q_{m/z,i}$  and  $W_i$  has been added in the last part of Section 2.3.

**P8, L214-215**

(11) P7L168-169: What causes the variability in sensitivity for certain ions with respect to the difference in the chemical form? Is this caused by differences in collection efficiency or vaporization efficiency/incomplete vaporization? The efficiency of electron ionization should be the same.

We have not identified the mechanism. A possible explanation would be variability in the divergence angle of evolved gas molecules after the thermal desorption. Uchida et al. (2019) and Ide et al. (2019) experimentally and theoretically showed that the divergence angle of

molecules could depend on the molecular weight. This point has been added in the last part of Section 3.1.

**P9, L233-237**

(12) P7L178: Is the relative peak area in the measurements proportional to the relative composition of the particles, i.e. can the composition of a multi-component particle be reliably calculated from single-component calibrations? This should be stated clearly.

Kobayashi et al. (2021) suggested that the ion signals for multi-component sulfate particles could be approximated as the linear combination of ion signals originating from single-component sulfate particles based on mass closure tests. We did not perform detailed mass closure tests in the current study because of significant uncertainties in determining the mass of multi-component particles (especially for the particles generated from the seawater samples). Alternatively, we compared the SN/SC and SS/SC ratios estimated from the QMS ion signals with those predicted from the ionic concentrations in the solutions. This point was added in the last part of Section 2.3.

**P7, L174-178**

(13) P8 Figure 2: Why is the NO+ signal in Figure 2(a) and (d) about 10 times more intense than the Na+/K+ signals (similar intensity after multiplication with 0.1)? Are Na and K incompletely vaporized?

This is related with the question (11). The sensitivities at m/z 30 for nitrate particles were larger than those for the other m/z peaks (Figs. 2 and 3). The difference in the sensitivities cannot be explained by the difference in the electron ionization cross sections. We have not identified the mechanisms that caused the variability in the sensitivity values. The difference in the divergence angle of evolved gas molecules after the thermal desorption might be a possible mechanism (Uchida et al., 2019; Ide et al., 2019). We have added this point in the last part of Section 3.1.

**P9, L233-237**

(14) P10 Figure 4: In Figure 4(b) the m/z 39 signal ( $K^+$ ) is strongly enhanced after the first group of peaks and even more after the second peak (40-50s). What causes this enhanced background signal? How do you deal with it when calculating the total signal area? Is this slowly vaporizing potassium?

The enhancement after the first group of peaks would be caused by the tailing of ion signals from PC particles and the onset of the thermal decomposition of PS particles. The enhancement after the second peak may be caused by the incomplete vaporization of PN particles. We have not investigated the effects of the enhanced background signals in the current study. This issue will be addressed in future studies.

**P9, L247-248**

**(15) P10L205: "pure seawater" should probably be "diluted seawater".Corrected. P14, L270-271**

(16) P12 Figure6: The y-axes captions "Molar ratio of Na2SO4/NaNO3 to NaCl in collected particles" should rather read "Molar ratio of Na2SO4/NaNO3 to NaCl from ion signal intensities" since the real ratio in the particles is not known (but probably is the same as in the solution) and the ion signal intensities are used for this comparison.

Corrected. P16, Fig. 6

(17) P13L243-249: These decomposition equations do explain the occurrence of the m/z 30 (NO+) signal, however, they do not explain the occurrence of the m/z 23 and 39 signals. Is the final product of these equations (Na2O2, K2O2) vaporized or is it further decomposed into Na/K and O2? Are these ions (Na2O2+, K2O2+) observed in the mass spectra?

(18) P13L250-251: I do not understand how the sequential thermal decomposition of the Nitrate salts causes the bimodal peaks at m/z 23 and 39. This would only be the case if the intermediate Na- and K-containing products would vaporize to form the Na+/K+ peaks. However, if this would be the case, why is not all the material vaporized during the first peak? Furthermore, if e.g. NaNO2 is vaporized, is the respective ion observed in the mass spectra?

Because the questions (17) and (18) are related with each other, we will collectively answer them. We have revised the explanation in Section 4.1. Our experimental data indicate that the thermal decomposition of SN particles yielded gas-phase Na and NO, and that of PN particles yielded gas-phase K and NO. However, the temporal evolution of the ion signals at m/z 23 and 39 suggests that the thermal decomposition processes of SN and PN particles were not represented by single-step reactions. Tagawa (1987) proposed the following reactions for thermal decomposition of bulk SN in dry air:

> $NaNO_3 \rightarrow NaNO_2 + 1/2 O_2 (\sim 491-750^{\circ}C)$   $NaNO_2 \rightarrow 1/2 Na_2O_2 + NO (\sim 750-850^{\circ}C)$  $Na_2O_2 \rightarrow Na_2O + 1/2 O_2 (> 850^{\circ}C)$

and those for PN in dry air:

$$\begin{array}{l} \text{KNO}_3 \rightarrow \text{KNO}_2 + 1/2 \text{ O}_2 \ (\sim 526 - 750^{\circ}\text{C}) \\ \text{KNO}_2 \rightarrow 1/2 \text{ K}_2\text{O}_2 + \text{NO} \ (\sim 750 - 900^{\circ}\text{C}) \\ \text{K}_2\text{O}_2 \rightarrow \text{K}_2\text{O} + 1/2 \text{ O}_2 \ (> 900^{\circ}\text{C}) \end{array}$$

We speculate that  $NaO_x$  and  $KO_x$  produced via the first step reactions underwent further thermal decomposition reactions to yield gas-phase Na and K in the rTDMS.

P17, L307-321

(19) P14Table3: Why are not the same collection times used in the LOD measurements as in

the other measurements of this study (i.e. 6 min instead of 2 or 4 min)?

The LODs are not critical for laboratory experiments but important for ambient measurements. We varied the collection time depending on the mass loadings. We tentatively used a 10-min measurement cycle (6-min collection time) for ambient test measurements at TMU (a suburban site). This cycle could be used under typical ambient conditions in urban air, except for very low mass loadings. Therefore, we set the collection time of 6 min for estimating the LODs. This point was briefly mentioned in the caption of Table 3.

**P18, Table 3**

(20) P14L292: What could these "matrix effects" be? What could cause these differences in signal intensities? How would mixtures of K- and Na-salts behave? These results show that Na cannot be quantitatively measured with this method. This should be stated clearly in Abstract and Conclusions.

We have added the description "The SS/SC ratios estimated from the ion signals at m/z 23, 36 (H35Cl+), and 48 (SO+) agreed well with those predicted from the solution concentrations to within ~10%. The SN/SC ratios estimated from the ion signals at m/z 30 (NO+) and 36 also agreed with those predicted from the solution concentrations to within ~15%, whereas the SN/SC ratios estimated from m/z 23 were significantly lower than the predicted values." in the abstract and conclusions.

The matrix effects mean that the sensitivities depend on coexisting compounds. We have added this point only in the conclusions because the descriptions in the abstract become too lengthy. We have not tested the mixtures of K- and Na-salt particles. The quantification of K-salt particles would be investigated in future studies.

P1, L21-25; P19, L375-376; P19, L389 - P20, L397

(21) P15L311: In this work it was not shown that the "current system achieved collection efficiencies of  $\sim$ 70% for solid sulfate particles". This was rather an assumption, based on previous measurements.

We have revised the sentence as "Kobayashi et al. (2021) showed that the collection efficiencies for laboratory-generated solid sulfate particles were  $\sim$ 70%." Please see the answer to the question (6) for the details of the collection efficiency.

**P20, L404-405**

(22) P15L315: Here, an alternative for the current ADL is discussed. Unfortunately, the authors do not mention the features or limitations of the current ADL in this paper.

We have added the information on the current ADL in Section 2.1 and 5. The structure of the ADL is the same as that used by Miyakawa et al. (2014), which is essentially identical to that presented by Zhang et al. (2004).

Other corrections:

- Fig. 3: We have corrected minor errors in calculating the mass of Cl (the isotopic fractions).
   We have also corrected errors in calculating the mass of K2SO4.
- Fig. 6: We have revised the y-axis values (the changes in the sensitivity for m/z 36 and modification of the multi-mode fitting). We have also updated the error bars.
- Table 3: The LOD value for PS has been slightly modified due to the change in the sensitivity.
- The data in the Supplement have also been updated.

Although these corrections do not alter the major conclusions, we apologize for the mistakes in the key results. We have also made minor corrections, adjustment of figure symbols (Fig. 4), and layout of tables in both the main document and Supplement.

**References:**

- Drewnick, F., Diesch, J.-M., Faber, P., and Borrmann, S.: Aerosol mass spectrometry: particle-vaporizer interactions and their consequences for the measurements, Atmos. Meas. Tech., 8, 3811-3830, https://doi.org/10.5194/amt-8-3811-2015, 2015.
- Kobayashi, Y., Ide, Y., and Takegawa, N.: Development of a novel particle mass spectrometer for online measurements of refractory sulfate aerosols, Aerosol Sci. Technol., 55, 371-386, https://doi.org/10.1080/02786826.2020.1852168, 2021.
- Matthew, B. M., Middlebrook, A. M., and Onasch, T. B.: Collection efficiencies in an Aerodyne Aerosol Mass Spectrometer as a function of particle phase for laboratory generated aerosols, Aerosol Sci. Technol., 42, 884–898, https://doi.org/10.1080/02786820802356797, 2008.
- Miyakawa, T., Takeda, N., Koizumi, K., Tabaru, M., Ozawa, Y., Hirayama, N., and Takegawa, N.: A new laser induced incandescence - mass spectrometric analyzer (LII-MS) for online measurement of aerosol composition classified by black carbon mixing state, Aerosol Sci. Technol., 48, 853-863, https://doi.org/10.1080/02786826.2014.937477, 2014.
- Robinson, E. S., Onasch, T. B., Worsnop, D., and Donahue, N. M.: Collection efficiency of α-pinene secondary organic aerosol particles explored via light-scattering single-particle aerosol mass spectrometry, Atmos. Meas. Tech., 10, 1139–1154, https://doi.org/10.5194/amt-10-1139-2017, 2017.
- Saleh, R., Robinson, E. S., Ahern, A. T., and Donahue, N. M.: Evaporation rate of particles in the vaporizer of the Aerodyne aerosol mass spectrometer. Aerosol Sci. Technol., 51, 501–508, https://doi.org/10.1080/02786826.2016.1271109, 2017.
- Zhang, X., Smith, K. A., Worsnop, D. R., Jimenez, J. L., Jayne, J. T., Kolb, C. E., Morris, J., and Davidovits,
  P.: Numerical characterization of particle beam collimation: Part II integrated aerodynamic-lens-nozzle system, Aerosol Sci. Technol, 38, 619–638, https://doi.org/10.1080/02786820490479833, 2004.

---

## Author Comment (AC2)

**Response to Referee 2**

This manuscript describes an analytical mass spectrometry-based method to measure refractory salts of sodium and potassium or atmospheric relevance. This work represents a useful advance for atmospheric measurements of these species, which are difficult or impossible to measure with current online instrumentation and often require slower, more labor-intensive offline analysis. The manuscript is well-written and straightforward, and I recommend it be published.

We would like to thank the referee very much for giving us valuable comments and suggestions. We have revised the manuscript to address those comments and also made other corrections to improve the clarity of the presentation. The line numbers for this response letter are based on the manuscript with track changes.

(1) My main suggestion is that the authors include some context for the stated goals of measuring potassium in ambient biomass burning particles. What is biomass burning K+ associated with? Can it be associated with organic compounds, and if so, what are the prospects for detecting it? On a similar note, potassium is present in biological aerosol (Christopher Pöhlker, Bärbel Sinha, Manabu Shiraiwa, et al., 2012). Should we expect that to be measured with similar efficiency to the compounds explored in this study?

The issue raised by the referee would be important in measuring biomass burning particles. KCl is the dominant form of chloride in biomass burning aerosols (Reid et al., 2005), whereas potassium may also be present in biological particles emitted from rainforest (Pöhlker et al., 2012). We have added these points in Section 1 and also added some of the suggested references. We consider that the maximum temperature of the graphite collector is sufficiently high to decompose potassium-containing organic particles. However, the current study was focused on the detection of inorganic salts, and we have not tested the quantification of organic particles. The quantification of biomass burning aerosols will be investigated in future studies.

P2, L38-40

(2) Line 90-91. The authors should note that atomizing seawater does not produce sea spray aerosol that has the same composition of real sea spray aerosol produced by wave breaking-driven bubble bursting, e.g. (Fuentes et al., 2010). The differences may be mostly due to the presence of organics, but association of organics with some major ions may result in differences in the major ions in sea spray, which likely would not be replicated in atomized samples, e.g. see Salter et al. (2016).

We thank the referee for useful information. The molar fractions of the inorganic salts in the laboratory-generated particles may be different from those in real-world sea salt particles. We have added this point in Section 2.2.

P4, L112-115

(3) Line 253. It is interesting that no NaCl cluster ions were observed. ClNa2+ is a singly charged ion that is observed in CIMS observations of thermally desorbed particles (Lawler et al., 2014). I think those observations show that NaCl can be desorbed as an intact molecule that can react with $Na^+$ to form the cluster. I wonder whether it is more a consequence of the ionization scheme (EI) that no cluster molecules were observed. Can the authors comment on this possibility?

The QMS was operated in the multiple ion detection mode to measure ion signals at some selected $m/z$ peaks. Before starting the full experiments, we measured all possible $m/z$ signals to identify the major $m/z$ peaks to be used for quantification (Table S1 in the Supplement). Following the referee's suggestion, we performed additional experiments to investigate possible contributions from the intact form of NaCl or its clusters. We found that ion signals at $m/z$ 58 ($Na^{35}Cl^+$) and 81 ($Na_2^{35}Cl^+$) were negligibly small as compared to those at $m/z$ 23 or 36, indicating that the contributions of the intact form of NaCl or its clusters were not significant.

P8, L195-197

(4) Table 3. It would be helpful to have the detection limits also reported in terms of total sample mass (ng).

We have added the detection limits in terms of total sample mass (ng) in Table 3.

P18, Table 3

(5) Figure 2. Can the authors comment on the wide gap in desorption time between the NaCl peaks and the $Na_2SO_4$ peaks? The bulk compounds differ in melting point by only about 80 K, while the difference in melting point of NaCl and $NaNO_3$ is several hundred degrees and the peaks for these two compounds show up close in time. Is the rate of temperature increase of the graphite cup strongly nonlinear or is there some other explanation? If it is possible to estimate the temperature over the heating period, that would of course be helpful to show.

The details of the laser power settings and the temperature of the graphite collector are presented in Section S2 in the Supplement. We consider that the issue raised by the referee is important. We used the two-step laser modulation to separately quantify the aerosol compounds tested in this study. While this method successfully separated SS from SN and SC, the ion signals for SN and SC showed temporal overlapping. This is somewhat unexpected considering that the difference in the bulk thermal desorption temperature between SN and SC is larger than that between SC and SS, as the referee pointed out. The cause of this discrepancy is currently unidentified. We have added this point in the last part of Section 4.1.

P17, L330-333

Other corrections:

- Fig. 3: We have corrected minor errors in calculating the mass of Cl (the isotopic fractions). We have also corrected errors in calculating the mass of $K_2SO_4$.
- Fig. 6: We have revised the y-axis values (the changes in the sensitivity for $m/z$ 36 and modification of the multi-mode fitting). We have also updated the error bars.
- Table 3: The LOD value for PS has been slightly modified due to the change in the sensitivity.
- The data in the Supplement have also been updated.

Although these corrections do not alter the major conclusions, we apologize for the mistakes in the key results. We have also made minor corrections, adjustment of figure symbols (Fig. 4), and layout of tables in both the main document and Supplement.

References:

Christopher Pöhlker, Bärbel Sinha, Manabu Shiraiwa, K. T. W., Sachin S. Gunthe Hang Su Paulo Artaxo Qi Chen Yafang, M. S., Cheng Mary K. Gilles Arthur L. D. Kilcoyne Ryan C. Moffet, W. E. and Markus Weigand Ulrich Pöschl Meinrat O. Andreae, S. T. M.: Biogenic Potassium Salt Particles as Seeds for Secondary Organic Aerosol in the Amazon, Science (80-. )., 337(August), doi:10.1126/science.1123264, 2012.

Fuentes, E., Coe, H., Green, D., de Leeuw, G. and McFiggans, G.: Laboratory-generated primary marine aerosol via bubble-bursting and atomization, Atmos. Meas. Tech., 3(1), 141–162, doi:10.5194/amt-3-141-2010, 2010.

Lawler, M. J., Whitehead, J., O'Dowd, C., Monahan, C., McFiggans, G. and Smith, J. N.: Composition of 15–85 nm particles in marine air, Atmos. Chem. Phys., 14(21), 11557–11569, doi:10.5194/acp-14-11557-2014, 2014.

Salter, M. E., Leck, C., Werner, J., Johnson, C. M., Riipinen, I., Nilsson, E. D. and Zieger, P.: Calcium enrichment in sea spray aerosol particles, Geophys. Res. Lett., 43, 8277–8285, doi:10.1002/2016GL070275.Abstract, 2016.

We have added some of the suggested references.

References for this reply letter:

Pöhlker, C., Wiedemann, K. T., Sinha, B., Shiraiwa, M., Gunthe, S. S., Smith, M., Su, H., Artaxo, P., Chen, Q., Cheng, T., Elbert, W., Gilles, M. K., Kilcoyne, A. L. D., Moffet, R. C., Weigand, M., Martin, S. T., Pöschl, U., and Andreae, M. O.: Biogenic potassium salt particles as seeds for secondary organic aerosol in the Amazon, Science, 337, 1075–1078, https://doi.org/10.1126/science.1223264, 2012.

Reid, J. S., Koppmann, R., Eck, T. F., and Eleuterio, D. P.: A review of biomass burning emissions part II: Intensive physical properties of biomass burning particles, Atmos. Chem. Phys., 5, 799–825, https://doi.org/10.5194/acp-5-799-2005, 2005.

---

## Author Response (AR2)

January 8, 2022

Dr. Pierre Herckes
Editor
Atmospheric Measurement Techniques

Manuscript title: "**A new method to quantify particulate sodium and potassium salts (nitrate, chloride, and sulfate) by thermal desorption aerosol mass spectrometry**" by Yuya Kobayashi and Nobuyuki Takegawa

Dear Dr. Herckes,

Thank you very much for handling our manuscript. I have uploaded the data to the Zenodo repository. I herewith certify that the content (e.g. text, figures, equations, or tables) of all files uploaded in this form is exactly the same as the version of my manuscript accepted by the Associate Editor, except for the Word file that incorporated the updated reference for the data repository and the Supplement file that excluded the data archived in the repository. Please see the manuscript with track changes for details.

Sincerely yours,

Nobuyuki Takegawa

[revised manuscript text omitted]